# Deep Reinforcement Learning with Plasticity Injection

**Evgenii Nikishin** [1,2]  **Junhyuk Oh** [1]  **Georg Ostrovski** [1]  **Clare Lyle** [1]
**Razvan Pascanu** [1]  **Will Dabney** [1]  **André Barreto** [1]

## Abstract

A growing body of evidence suggests that neural networks employed in deep reinforcement learning (RL) gradually lose their plasticity, the ability to learn from new data; however, the analysis and mitigation of this phenomenon is hampered by the complex relationship between plasticity, exploration, and performance in RL. This paper introduces *plasticity injection*, a minimalistic intervention that increases the network plasticity without changing the number of trainable parameters or biasing the predictions. The applications of this intervention are two-fold: first, as a diagnostic tool — if injection increases the performance, we may conclude that an agent's network was losing its plasticity. This tool allows us to identify a subset of Atari environments where the lack of plasticity causes performance plateaus, motivating future studies on understanding and combating plasticity loss. Second, plasticity injection can be used to improve the computational efficiency of RL training if the agent has to re-learn from scratch due to exhausted plasticity or by growing the agent's network dynamically without compromising performance. The results on Atari show that plasticity injection attains stronger performance compared to alternative methods while being computationally efficient.

## 1. Introduction

"You cannot teach an old dog new tricks" an old proverb says. While the common wisdom is not necessarily a source of ground truth, neuroscientists recognized a long time ago that biological agents indeed gradually lose adaptability with age (Livingston, 1966). This phenomenon is referred to as loss of plasticity in brains (Nelson, 1999; Mateos-Aparicio

& Rodríguez-Moreno, 2019) and happens for multiple reasons, including natural degradation of neurons and their connections (Mahncke et al., 2006; Kolb & Gibb, 2011).

Since the biological causes for loss of plasticity do not apply to artificial agents, in principle there is no reason to expect that this phenomenon also happens in the context of machine learning. Surprisingly, several recent works show that reinforcement learning (RL) agents that use neural networks may gradually lose the ability to learn from new experiences (Dohare et al., 2021; Lyle et al., 2022; Nikishin et al., 2022).

The precise mechanisms causing loss of plasticity in RL are not well understood. The problem is particularly challenging to study in this context because performance in RL is influenced by many factors. For example, an agent without plasticity issues may still struggle to learn if it fails to properly explore the environment (Taïga et al., 2019). Past literature focused on using and controlling proxy measures of plasticity such as the number of saturated rectified linear units (Nair & Hinton, 2010) and feature rank (Kumar et al., 2021), but it is unclear how well these measures manage to capture the underlying phenomenon (Gulcehre et al., 2022).

This paper complements past evidence about the existence of plasticity loss in deep RL and introduces *plasticity injection*, an intervention that augments plasticity of the agent's neural network. The conceptual idea is simple: at any point in training, one can freeze the current network and create a new one that is going to be learning a change to the predictions, whilst ensuring that the change is initially zero. Crucially, plasticity injection does not increase the number of *trainable* parameters and does not affect the network's predictions when it is applied. Because of these properties, the intervention enables careful analysis of the plasticity loss phenomenon in RL while keeping other confounding factors aside.

We suggest two uses of plasticity injection, one as an analytic tool and another as a practical algorithmic technique. For analysis, we propose an experimental protocol that uses plasticity injection for *diagnosing the problem* of plasticity loss: for example, if an agent that was struggling to improve its behavior escapes a performance plateau after the intervention, we can conclude that the agent had been ex-

---

[1]DeepMind [2]Work done during the internship; currently at Mila, Université de Montréal. Correspondence to: Evgenii Nikishin <evgenii.nikishin@mila.quebec>.

*Reincarnating RL Workshop at the ICLR Conference*, Kigali, Rwanda, 2023. Copyright 2023 by the author(s).

periencing problems with its network plasticity. Using this protocol in the Arcade Learning Environment (Bellemare et al., 2013), we identify scenarios where loss of plasticity hinders the learning process. Furthermore, based on the intervention-enabled analysis, we provide recommendations for controlling the degree of plasticity loss.

We also propose to use plasticity injection as a way to improve computational efficiency of RL training in the following scenarios. First, when the agent loses its plasticity because its network turns out to be too small, plasticity injection can be dynamically used to increase the capacity of the agent without having to re-train the agent with a larger network from scratch. We empirically show that our method improves the aggregate score across 57 Atari games by 20% compared to other methods for dynamically addressing plasticity loss. Second, plasticity injection can be used to minimize computation by switching from a small network to a larger network in the middle of training without compromising the performance compared to using the larger network from scratch; we also empirically verify it on Atari games.

To summarize, our contributions include:

1. A minimalistic intervention called *plasticity injection* that increases plasticity of the agent while preserving the number of trainable parameters and not affecting its predictions;

2. Complementary evidence about the existence of the loss of plasticity phenomenon in deep RL;

3. An experimental protocol for diagnosing plasticity loss using the intervention;

4. A way to improve computational efficiency of RL training by dynamically expanding the network.

## 2. Related Work

**Plasticity in Continual Learning.** Discussions about plasticity of neural networks date back (at least) to a seminal paper by McCloskey & Cohen (1989) outlining the plasticity-stability dilemma, a trade-off between preserving performance on previous tasks and maintaining adaptability to future ones. The continual learning community historically put a higher emphasis on the stability aspect, addressing catastrophic forgetting of past behaviors (French, 1999). Recently, several works raised awareness of difficulties with learning on future tasks too. Ash & Adams (2020) demonstrated an instance of loss of generalization, when pre-training a network might unrecoverably damage generalization even if pre-training was done on a uniform subsample of the same dataset. Berariu et al. (2021) deepened the study and conjectured that the phenomenon might

happen because of the reduction of gradient noise when warm-starting the network. Dohare et al. (2021) explicitly study the network plasticity in continual learning and demonstrate the reduced ability to minimize even the training error as the number of tasks increase. These works build an understanding of the problem by studying simplified settings that isolate different aspects of learning capabilities in continual learning, whereas our work aims at tackling the deep RL setting in its whole.

**Loss of Plasticity in Deep RL.** Issues with plasticity and related phenomena have been recently highlighted in deep RL under a plethora of different names. Lyle et al. (2022) show loss of capacity for fitting targets in online RL and Kumar et al. (2021) demonstrate a related implicit under-parameterization phenomenon caused by bootstrapping with more emphasis on the offline RL case. Both of these works use the feature rank as a proxy measure for plasticity but later Gulcehre et al. (2022) question the reliance on such measure by demonstrating a weak correlation between the rank and the agent's performance, partially motivating our study that focuses directly on agent's performance to reason about plasticity. Works of Sokar et al. (2023); Abbas et al. (2023) focus on saturation of neurons over the course of training, but Lyle et al. (2023) demonstrate that the saturation alone cannot fully characterize the plasticity loss phenomenon. Nikishin et al. (2022) discuss the primacy bias in deep RL, a tendency to excessively train on early data damaging further learning progress, and propose to periodically reset a part of the network to address the issue while relying on the replay buffer as a knowledge transfer mechanism. Earlier, Igl et al. (2021) had observed that deep RL agents can lose the ability to generalize due to non-stationarity and proposed to use distillation as a mitigation mechanism. Plasticity injection closely relates to these approaches by leveraging newly initialized weights, but does not require re-training and directly continues learning.

**Architectures.** The works above mainly discuss algorithmic aspects with less focus on the network architecture, although it is also an important component of the agent's design (Mirzadeh et al., 2022). The closest work in this space is about progressive networks (Rusu et al., 2016) that considers a setting with multiple environments and adds a new network with cross-connections to the layers of previous networks. A network after plasticity injection can be viewed as a simplified version of the architecture with a motivation of increasing plasticity within a single task without affecting agent's predictions. A line of work on the mixture of experts (Shazeer et al., 2017) and modular networks (Andreas et al., 2016) is also related, but typically focus in these papers is compositionality or handling multi-modalities. The idea of growing network layers or neurons has also been investigated (Fahlman & Lebiere, 1989; Chen et al., 2015); plasticity injection belongs to a family of these

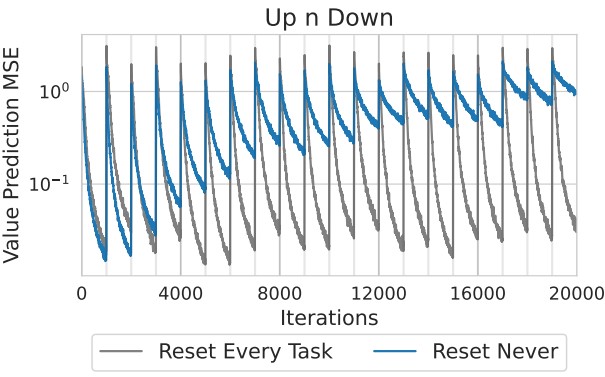

Figure 1. Demonstration of plasticity loss in a sequence of policy evaluation tasks. The task (a policy to evaluate) changes every 1000 iterations. The *reset every task* setting shows that newly-initialized parameters are able to fit each task, whereas the *reset never* setting shows the diminishing capability to fit the data when using the trained parameters from one task as the initialization for another task.

methods up to the difference that we explicitly control for maintaining the number of trainable parameters. In the context of language modeling, Hu et al. (2022) explored a similar idea of freezing a pre-trained model and fine-tuning a low-rank addition to the weight matrices on a downstream task. Lastly, plasticity injection can be conceptually viewed as an instance of residual learning (He et al., 2016) and boosting (Schapire, 1990).

## 3. An Illustration of Plasticity Loss

Plasticity of a neural network is broadly defined as the ability to learn from new experiences. To provide intuition on how this ability can decrease over time, we present a didactic example before investigating the case with deep RL. Figure 1 shows the mean-squared error (MSE) on a sequence of supervised policy evaluation problems derived from the Up n Down Atari environment. We first trained an agent on this environment for 200M frames and stored the policies occurring at every 10M frames. Then, for each stored policy, we sampled states from the corresponding stationary distribution and computed Monte-Carlo estimates of the value function for each state, resulting a training set composed of states and the values. We then trained a network to solve the resulting sequence of prediction problems. This sequence of related prediction problems differing in the input and target distribution aims to reproduce the scenario faced by an online RL agent (Dabney et al., 2021). The curve labeled "reset never" corresponds to starting each prediction problem using the final parameters from the previous one, while "reset every task" corresponds to randomly initializing the network parameters at every prediction problem.

The conventional wisdom about transfer learning suggests that, if two tasks are related, pre-training on the first might accelerate learning on the second (Pan & Yang, 2009). Here we observe the opposite trend: it takes longer and longer for the network to decrease training error on the subsequent policy evaluation problems if its parameters are not re-initialized. This example gives a simple demonstration of how plasticity loss can occur; we refer to the work by Dohare et al. (2021) for an in-depth study of the phenomenon in the continual setting.

After building the intuition about loss of plasticity, we turn the attention to its analysis in deep RL. The key distinctive feature of RL is the presence of an exploration confounder: in contrast to the continual setting with a fixed sequence of datasets, an RL agent influences the future data it learns from. Thus, a failure of an RL system can be attributed not only to loss of plasticity but also to inability to explore. The next section presents a strategy to increase plasticity of an agent that addresses the difficulty with the analysis.

## 4. Plasticity Injection

Before describing the experimental design in detail, we list the motivating desiderata:

- **Unaffected predictions:** the agent's predictions should stay the same after the intervention to avoid abrupt changes. This criterion allows isolating confounding factors related to exploration;

- **Preserving the trainable parameter count:** the intervention should not affect the number of trainable parameters to minimize confounding factors from an increased representational capacity.

We now present the proposed intervention to increase plasticity of an RL agent. First, let us denote the neural network approximator employed by the agent (for example, used for action-value prediction) as $h_\theta(x)$, where $\theta$ indicates the parameters. At some point in training, where the network might have started losing plasticity, we are going to freeze the parameters $\theta$ and introduce a new set of parameters $\theta'$ sampled from random initialization. The key idea is to keep two copies of $\theta'$, which we denote by $\theta'_1$ and $\theta'_2$; while $\theta'_1$ are free parameters used to learn a residual to the old network outputs, $\theta'_2$ remains frozen throughout. The agent's predictions after plasticity injection will be calculated using the following expression:

$$\underbrace{h_\theta(x)}_{\text{frozen}} + \underbrace{h_{\theta'_1}(x)}_{\text{trained}} - \underbrace{h_{\theta'_2}(x)}_{\text{frozen}}. \tag{1}$$

Since initially $\theta'_1 = \theta'_2$, immediately after plasticity injection the predictions of the neural network remain un-

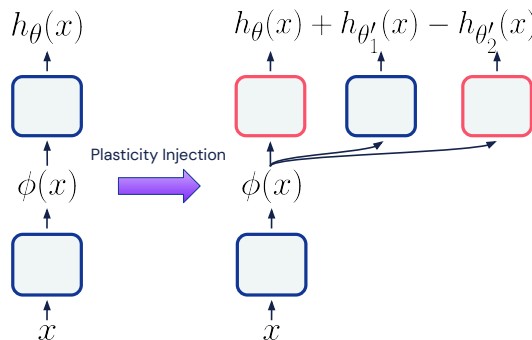

$$h_\theta(x) \qquad\qquad h_\theta(x) + h_{\theta'_1}(x) - h_{\theta'_2}(x)$$

Plasticity Injection

$$\phi(x) \qquad\qquad\qquad \phi(x)$$

$$x \qquad\qquad\qquad\qquad x$$

*Figure 2.* An illustration of the architecture before and after plasticity injection. Before the intervention, the network is schematically separated into an encoder $\phi(\cdot)$ and a head $h_\theta(\cdot)$, both parts are learning. After plasticity injection, we freeze the parameters $\theta$ of the head (we use red to indicate parameters that are not updated in the illustration) and create two copies of a randomly initialized parameters $\theta'$; one frozen and one unfrozen. The output of the agent is obtained by first passing the input $x$ to the encoder $\phi(\cdot)$, next passing $\phi(x)$ to all three heads, and finally combining the heads' outputs according to Expression (1).

altered. As learning progresses, $\theta'_1$ deviates from $\theta'_2$ and $h_\theta(x) - h_{\theta'_2}(x)$ serves as a bias term for predictions.

Note that if we apply plasticity injection to all parameters of the network, the new network will have to re-learn the representations encoded in $h_\theta(\cdot)$ from scratch. Thus, we apply our intervention to only a subset of the parameters and explain the idea further with a slight abuse of notation. We schematically split the network into an *encoder* $\phi(\cdot)$, that denotes a mapping induced by first $k$ layers of the network, and a *head* $h_\theta(\cdot)$ where $\theta$ now refers to parameters of the remaining layers of the network. After this relabelling, we can apply the intervention to $h_\theta(\cdot)$ as outlined above. Section 5.4 later presents an ablation of sharing the encoder.

Figure 2 illustrates the strategy to apply plasticity injection. Note that gradients from the frozen heads affect the encoder too, i.e. we do not stop the gradient propagation from any of the components of the output. It is worth noting that the proposed intervention increases the total number of parameters of the network (but keeps the same number of *trainable* parameters), which in turn may result in an increase of training time. However, we later discuss in Section 5.3 how plasticity injection can *save* computational resources.

The idea of learning with newly-initialized last layers has been explored by Nikishin et al. (2022), who suggested resetting the corresponding parameters of the network at fixed intervals and used the replay buffer (Lin, 1992) to re-learn after resets. Their experimental evidence supports the hypothesis that resets mitigate plasticity loss. Note though that resetting parameters of the network abruptly changes its

predictions, which results in a temporary decrease in performance and induces an exploration effect. From an analysis perspective, these abrupt changes make it more difficult to isolate the effect of additional plasticity on the agent's performance. From a practical perspective, plasticity injection does not rely on the buffer; Section 5.3 demonstrates how this difference can be critical.

## 5. Experiments

This section presents results for two main applications of plasticity injection: as a tool for diagnosing plasticity loss and as a way to dynamically grow the network to efficiently use computations. Afterwards, we demonstrate detailed ablations on the design choices when using plasticity injection.

### 5.1. Experimental Setup

The baseline agent is Double DQN (Van Hasselt et al., 2016) learning for 200M interactions on a standard set of 57 Atari games from the Arcade Learning Environment benchmark (Bellemare et al., 2013). The choice of Double DQN is motivated by the relative robustness and stronger performance of the agent with double Q-learning (Van Hasselt, 2010) compared to the vanilla DQN agent (Mnih et al., 2015) as well as simplicity compared to later DQN-based agents such as Rainbow (Hessel et al., 2018).

The majority of the experiments use a single plasticity injection after 50M frames; otherwise, we explicitly specify the number and timesteps of injections. A convolutional neural network employed by the Double DQN agent consists of 5 layers. The encoder corresponds to the first three of them (hence $k = 3$), while the head refers to the last two. Since DQN-based agents employ a target copy of the network parameters, we perform the same interventions on them.

For reliable evaluation of the aggregate performance across environments, we adopt the protocol of Agarwal et al. (2021) with a focus on the interquartile mean (IQM). All experiments use 3 random seeds.

### 5.2. Plasticity Injection as a Diagnostic Tool

Consider the task of improving a deep RL system when an agent performs suboptimally. Practitioners know how non-trivial is the process of pinpointing exact reasons why an agent might be struggling to improve the behavior. One of the reasons, as we discussed, can be loss of network plasticity throughout training.

We view the proposed intervention as a tool that can provide insight when analyzing deep RL systems. The procedure for using it is as follows: when an agent has a performance plateau or slower learning progress, take a saved copy of the agent, perform plasticity injection, and compare the

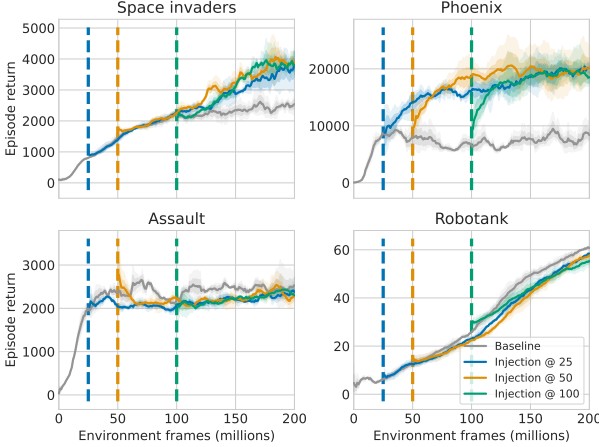

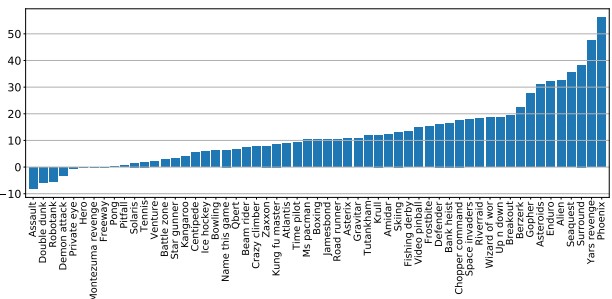

*Figure 4.* Percentage improvement of the average performance after adding plasticity injection across all 57 Atari games. We take the maximum score among the agents with plasticity injection after 25M, 50M, and 100M steps to roughly estimate the improvement as if plasticity injection was applied at a proper timestep and to demonstrate what the performance could have been if plasticity loss was mitigated. Learning curves corresponding to each environment are available in Appendix A.

*Figure 3.* A demonstration of diverse effects from plasticity injection applied to Double DQN after 25M, 50M, and 100M steps on a selection of Atari games comprising two examples where the intervention improves the performance and two examples where it does not. The baseline in `Space invaders` and `Phoenix` demonstrates the diminishing performance improvements and the performance plateau respectively, whilst the agent after the injection is capable of achieving higher returns. The stalled performance in `Assault` is due to exploration challenges (see Appendix F for details): adding plasticity could not alleviate them. If the agent does not show signs of the diminished ability to learn, like in `Robotank`, the injection would not lead to improved performance. Varying the injection timestep allows identifying the moment plasticity loss occurs. Results for all 57 environments are available in Figure 9.

training curves with and without the intervention. This way we answer a counterfactual question: what could have been the agent's performance if the network had more plasticity?

Figure 3 gives a set of example behaviors after following the procedure: in `Space invaders`, the baseline agent keeps learning but the post-injection agent improves at a faster rate towards the end of learning; we might interpret the observation as an indication of decreasing network plasticity over the course of training. In `Phoenix`, we see a completely stalled performance and the intervention allows doubling the final returns; such an observation point at possible *catastrophic* loss of plasticity, where additional interactions do not translate to better behavior. In `Assault`, on the other hand, the agent has plateaued but the injection does not make a difference. Further inspection revealed that around a score of 2800, the environment transitions to a new regime where an agent needs to start using an action that was not relevant before (see Appendix F for a visualization). This observation suggests that performance stagnation is related to exploration. In `Robotank`, the learning progress shows no signs of pathologies, giving evidence that the agent does not experience problems with its plasticity.

Plasticity injection can also demonstrate *when* loss of plasticity occurs. The post-intervention performance in `Space invaders` does not differ for varying injection timestep, suggesting that the agent might not start experiencing consequences of the lost plasticity until around 100M frames. On the other hand, in `Phoenix`, plasticity injection improves the performance earlier, implying that the agent lost its plasticity around 25M frames. Varying the moment of injection in `Assault` and `Robotank` does not change the performance significantly, supporting our previous conclusion about these games.

Figure 4 summarizes when and to which extent the Double DQN agent benefits from plasticity injection across 57 Atari games. The observations about improvements from injection complement evidence of the existence of plasticity loss in deep RL (Kumar et al., 2021; Lyle et al., 2022; Nikishin et al., 2022). We note that the argument here is nuanced: since the notion of plasticity is defined broadly and is challenging to measure, *it is our best interpretation* that the post-intervention agent can learn further because it addressed plasticity issues. But because of an experimental design that strived to be careful, we believe that it is the most likely explanation.

What should we do after using the tool and observing loss of plasticity? Dohare et al. (2021); Nikishin et al. (2022); Gogianu et al. (2021) provide evidence that the learning rate (LR), the replay ratio (RR)[1], the network size, and normalizations (such as spectral norm (SN) (Miyato et al., 2018)) strongly affect plasticity loss. We measure the sensitivity of the aggregate improvements of the final score from plasticity injection at 50M with respect to these choices of the

---

[1]The replay ratio denotes the number of gradient steps per an environment step.

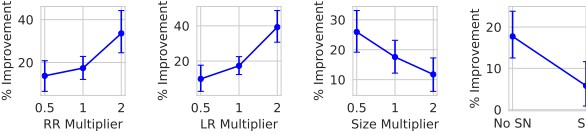

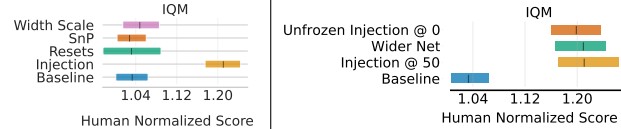

*Figure 5.* Percentage improvements of the IQM scores from plasticity injection in varying regimes controlling the degree of plasticity loss. The intervention effect size monotonically increases with the replay ratio (RR) and the learning rate (LR), monotonically decreases with the size of the neural network, and is smaller yet positive for an agent employing spectral normalization (SN). These observations can be seen as recommendations about how to address loss of plasticity.

agent specification[2]. Results in Figure 5 are consistent with observations from previous works and suggest a recipe for controlling the degree of plasticity loss by decreasing the learning rate or the replay ratio, increasing the network size, or employing normalizations[3].

In addition to gaining scientific insight, we now discuss how the intervention can be useful in large-scale RL.

### 5.3. Plasticity Injection for Computational Efficiency

Over the recent years, RL agents have been trained at increasingly larger scales. For example, mastering particularly challenging environments required an equivalent of hundreds of years of human gameplay (Vinyals et al., 2019), or obtaining a diverse set of skills required a 1B+ parameter networks (Reed et al., 2022). Given the trend, computational considerations become increasingly relevant.

Plasticity injection can be used to improve the computational efficiency of RL in the following ways.

**Reincarnating with Plasticity Injection.** Recently, Agarwal et al. (2022) proposed a workflow called "Reincarnating RL" which reuses computations from previously trained agents during the iterative process of agent design. For example, if we trained an agent for several days or weeks and then decided to change its design (such as the network size), the workflow suggests to leverage the spent computations instead of training again from scratch. Plasticity injection can be useful from this perspective when the agent is unable to improve due to loss of plasticity and re-training from scratch is expensive. To see the effectiveness of plasticity injection

---

[2]To make a network two times larger, we multiply the width of all hidden layers by $\sqrt{2}$.

[3]We follow the recommendation of Gogianu et al. (2021) and apply SN to the penultimate layer; since we apply injection to the last two layers, issues with their plasticity might be partially alleviated by SN. Given that Gogianu et al. (2021) notice that the spectral norm of other layers starts growing more and D'Oro et al. (2023) observe that the first layers benefit from partial resets, we conjecture that first layers' plasticity is still declining with SN.

*Figure 6.* **Left:** Comparison of plasticity injection to other methods that can be applied to dynamically address loss of plasticity. The difference in performance between all methods and the baseline is insignificant except for injection; an agent with injection is capable of improving without having to re-train from scratch. **Right:** Comparison of the agent with plasticity injection to agents that use larger networks from the beginning. *Injection @ 50* switches from a small network to a larger network through plasticity injection at 50M frames. *Unfrozen Injection @ 0* uses the same network as the agent with plasticity injection, but from the beginning of training, and updates both $\theta$ and $\theta_1'$ parameters. *Wider Net* uses a network with an increased width layers. *Injection @ 50* achieves higher or similar performance while saving computational resources during the first 50M frames.

in such setting, we compared plasticity injection with several alternatives that address loss of plasticity dynamically during training, including Shrink-and-Perturb (SnP) (Ash & Adams, 2020), resets (Nikishin et al., 2022), and naive width scaling (we describe the methods in detail in Appendix D). Figure 6 (left) shows that plasticity injection achieves a higher aggregate score across 57 Atari games compared to the alternatives. The results suggest that plasticity injection can be used to "reincarnate" agents more efficiently compared to the alternatives, without re-training from scratch.

**Minimizing Computations via Dynamic Growth.** Although larger networks tend to maintain plasticity longer, they require more computations to train or can be more challenging to train (Team et al., 2023). We hypothesize that the full capacity of a large network may not always be necessary early in training, even if it is useful to maintain plasticity later. If this hypothesis is true, we can save computations by starting from a smaller network and injecting plasticity during training, without compromising the final performance compared to using the large network from the beginning. To verify this hypothesis, we implemented two additional baselines with higher-capacity networks. The first uses a network with larger width layers, roughly matching the total number of parameters in $\phi(\cdot)$, $h_\theta(\cdot)$, and $h_{\theta_1}(\cdot)$ combined. The second uses the same network as an agent after plasticity injection but performs the intervention from the start and keeps $h_\theta(\cdot)$ unfrozen. In other words, the latter baseline uses a large network from the beginning of training, while the agent with standard plasticity injection starts from a smaller network and switches to the large network during training. The results in Figure 6 (right) show that an agent with plasticity injection during training performs comparably to the alternatives that use larger networks from the

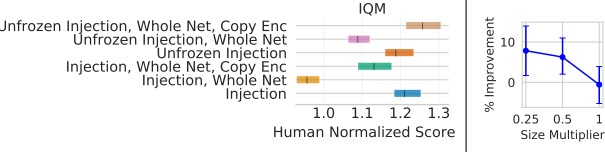

*Figure 7.* **Left:** Comparison between variations of plasticity injection. *Whole Net* denotes injection of both the encoder $\phi(\cdot)$ and the head $h_\theta(x)$; *Copy Enc* denotes copying the $\phi(\cdot)$ at the moment of injection without further sharing; *Unfrozen* denotes keeping parameters $\theta$ of the first term, $h_\theta(x)$, unfrozen. Relying on a new encoder leads to a lower performance; the rest of the alternatives have comparable scores. **Right:** Percentage improvements of the IQM score from multiple injections over a single injection for varying network sizes. Multiple injections are beneficial for smaller networks. Note that previous plots in Figure 5 show improvements when comparing one injection over no injections while this plot compares multiple injections over one.

start. At the same time, it saves computations since it uses a smaller network up to 50M frames and has fewer parameters that are updated even after plasticity injection. Appendix E provides a calculation of the amount of resources plasticity injection has saved during these experiments. These results confirm the hypothesis and suggest that plasticity injection can be used as a tool for minimizing computations when training RL systems at a large scale.

### 5.4. Ablations

This section presents an ablation analysis of the various design choices made during the study of plasticity injection. The purpose of such ablations is to build intuition on the behavior of plasticity injection under different conditions so that an RL practitioner can use it in their application.

**Injection Variants.** The proposed modification of the network architecture is not the only one possible. In Section 4, we initially described a version of plasticity injection without encoder sharing, that is, when the intervention is applied to the entire network (referred to as *Injection, Whole Net* in Figure 7). Another alternative is to create a whole new set of parameters and copy the encoder parameters of the old network without sharing it (denoted as *Injection, Whole Net, Copy Enc*). Lastly, for all three versions, there is the possibility of *not* freezing the old set of parameters (weights corresponding to the third, output correction term $h_{\theta_2'}(x)$, are always going to be frozen).

Figure 7 (left) summarizes the findings:

1. Creating a completely new encoder-head pair is the alternative with the lowest IQM scores;

2. Variants with encoder sharing or copying have comparable performance; the *Injection, Whole Net, Copy Enc*

version has a slightly lower performance than the rest. We conjecture that it might be due to the larger number of frozen parameters;

3. Unfrozen variants generally perform not worse than their frozen counterparts. The unfrozen variants introduce more trainable parameters compared to the baseline, which require more computations during learning and increase the network expressivity. Since we were interested in a careful diagnosis of plasticity loss and extra expressivity may be a confounding factor, we decided to stick to the frozen version by default.

**Multiple Injections.** Given the improved performance from plasticity injection in the previous experiments, a natural question is whether applying plasticity injection multiple times would improve performance even further. To investigate this question, we applied plasticity injection at 100M and 150M frames, in addition to 50M frames, and plotted the IQM improvements with respect to a single injection at 50M frames. As shown in Figure 7 (right), additional injections do not improve the performance over a single injection in a setup with a standard network. We hypothesize that in our particular experimental setting, loss of plasticity can be largely mitigated with a single plasticity injection. To verify this hypothesis, we applied multiple injections while varying the network size[4]. Figure 7 (right) confirms that the level of improvement grows monotonically as the agent uses smaller networks. Since the results in Figure 5 suggests that the degree of plasticity loss increases with smaller networks, this result indicates that multiple rounds of plasticity injection can be beneficial in situations where the agent network is too small to maintain plasticity.

**No Output Correction.** In the majority of the games, subtracting the initial copy of the newly introduced head $h_{\theta_2}(\cdot)$ resulted in mostly similar learning curves as without the subtraction, although not always. In particular, the impact of the injection on `Yars Revenge` is smaller without compensating for the bias. Also, we observed a significant difference in high variance games (such as `Berzerk` and `Hero`). Note that minimizing effects on the predictions from introducing the new head would be possible by modifying the initialization scheme (Brohan et al., 2022). We highlight that the goal was to have an as clean and simple experimental design as possible: the correction offered by the $h_{\theta_2'}(\cdot)$ term *guarantees* no effect on predictions; without it, effects can be initialization or domain specific.

**Injection Timestep.** Earlier in Section 5.2, we presented the results for a selection of environments for varying injection timestep. Figure 8 (left) suggests that across all games, increasing or decreasing the timestep by a factor of two

---

[4]Similarly to Section 5.2, to make the network 2x smaller, we divide the width of the hidden layers by $\sqrt{2}$.

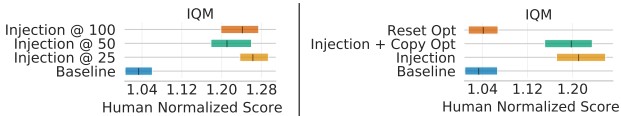

*Figure 8.* **Left:** Aggregate performance for agents with varying injection timesteps. Whilst Figures 9 and 10 suggest that loss of plasticity might be happening at different paces across environments, the final IQM score is relatively robust with respect to the injection moment. **Right:** Comparison of an agent with injection, an agent with injection but copied optimizer state for the newly initialized head (*Injection + Copy Opt*), and an agent that resets the optimizer statistics of the last two layers (*Reset Opt*). The results suggest that effects from interventions on the optimizer state are marginal compared to having new weights.

yields comparable aggregate performance. Note though that we measure the IQM score after 200M frames, so the transient performance would differ depending on the timestep. Appendix B also discusses later an adaptive criterion for choosing the injection timestep.

**Optimizer.** One might hypothesize that benefits from injection can be attributed to manipulations with the optimizer state. To test whether this hypothesis, we perform two ablations: the first resets statistics of the RMSProp optimizer (Tieleman et al., 2012) used by Double DQN after 50M steps, the second copies the optimizer state of the original head to the newly initialized head after the injection. Figure 8 (right) demonstrates that most of the effects from injection come from having additional weights rather than from interventions on the optimizer.

## 6. Limitations

The first and foremost limitation of plasticity injection is an increase in memory and training time. When using plasticity injection as a diagnostic tool, we believe the overhead is largely justified since preserving the network output makes it easier to isolate confounding factors like exploration. From the deployment viewpoint, the increase in compute and time may or may not be justified depending on how much plasticity injection improves performance. In our experiments, the effect of the intervention varied considerably across Atari games: while in some cases it did not help much, in other cases it had a significant positive effect.

Preserving the network outputs and keeping weights can be undesirable in case of a parameter divergence that often occurs in deep RL experimentation (Van Hasselt et al., 2018). Such a scenario also qualifies as loss of plasticity; in this case, addressing it without drastic tools can be challenging. Lastly, while we propose a diagnostic and mitigation tool, we do not identify causal factors driving plasticity loss in deep RL. More research is needed here: understanding these causes could lead to avoiding plasticity loss in the first place.

## 7. Discussion and Conclusion

Results in this paper can serve as a clear study of the plasticity loss phenomenon in deep RL and evidence that the RL optimization still leaves room for improvement. The version of plasticity injection we propose may yet not be optimal: we strived for simplicity rather than performance and view the intervention as a blueprint for future methods.

The experiments in this paper adopted the convolutional architecture from Van Hasselt et al. (2016) but modern deep RL practice not rarely involves ResNets (He et al., 2016; Espeholt et al., 2018) and Transformers (Vaswani et al., 2017; Chen et al., 2021; Reed et al., 2022); we did not investigate settings with these advanced architectures. However, the idea of plasticity injection is agnostic to the choice of the architecture. For example, it can be applied for residual blocks in ResNets or decoder blocks in Transformers.

An exciting avenue for future research is understanding trade-offs between architectural design decisions: RL agents typically employ networks that were originally proposed for stationary problems, but perhaps dynamically growing networks would suit the non-stationary nature of RL better.

Applications of plasticity injection focus on diagnosing RL systems and their efficiency. We compliment a recent opinion paper from Mannor & Tamar (2023) by arguing that if deep RL is to become a technology that a non-expert can use, more research is needed on the process of iterating on the agent design and computational efficiency.

Although this paper attempted to understand and address loss of plasticity in RL, there are still remaining open questions. Can we solve the problem of plasticity loss completely? Which properties of newly initialized networks enable high plasticity? Answering these questions is a key challenge for training truly intelligent agents.

## Acknowledgements

EN thanks Tom Schaul, Greg Farquhar, John Quan, Dan Horgan, Mihaela Rosca, Angelos Filos, Diana Borsa, Luisa Zintgraf, Yash Chandak, Robert Lange, Chris Lu, David Parkes, Blanca Huergo, Rich Sutton, David Silver, Hado van Hasselt, Alexander Novikov, Julia Novikova, and especially Iurii Kemaev for their help and valuable discussions. EN also thanks many other interns and the broader DeepMind team for the great internship experience.

We acknowledge the Python community (Van Rossum & Drake Jr, 1995; Oliphant, 2007) for developing the core set of tools that enabled this work, including JAX (Bradbury et al., 2018; Babuschkin et al., 2020), Jupyter (Kluyver et al., 2016), NumPy (Oliphant, 2006; Van Der Walt et al., 2011), SciPy (Jones et al., 2014), Matplotlib (Hunter, 2007), and pandas (McKinney, 2012).

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

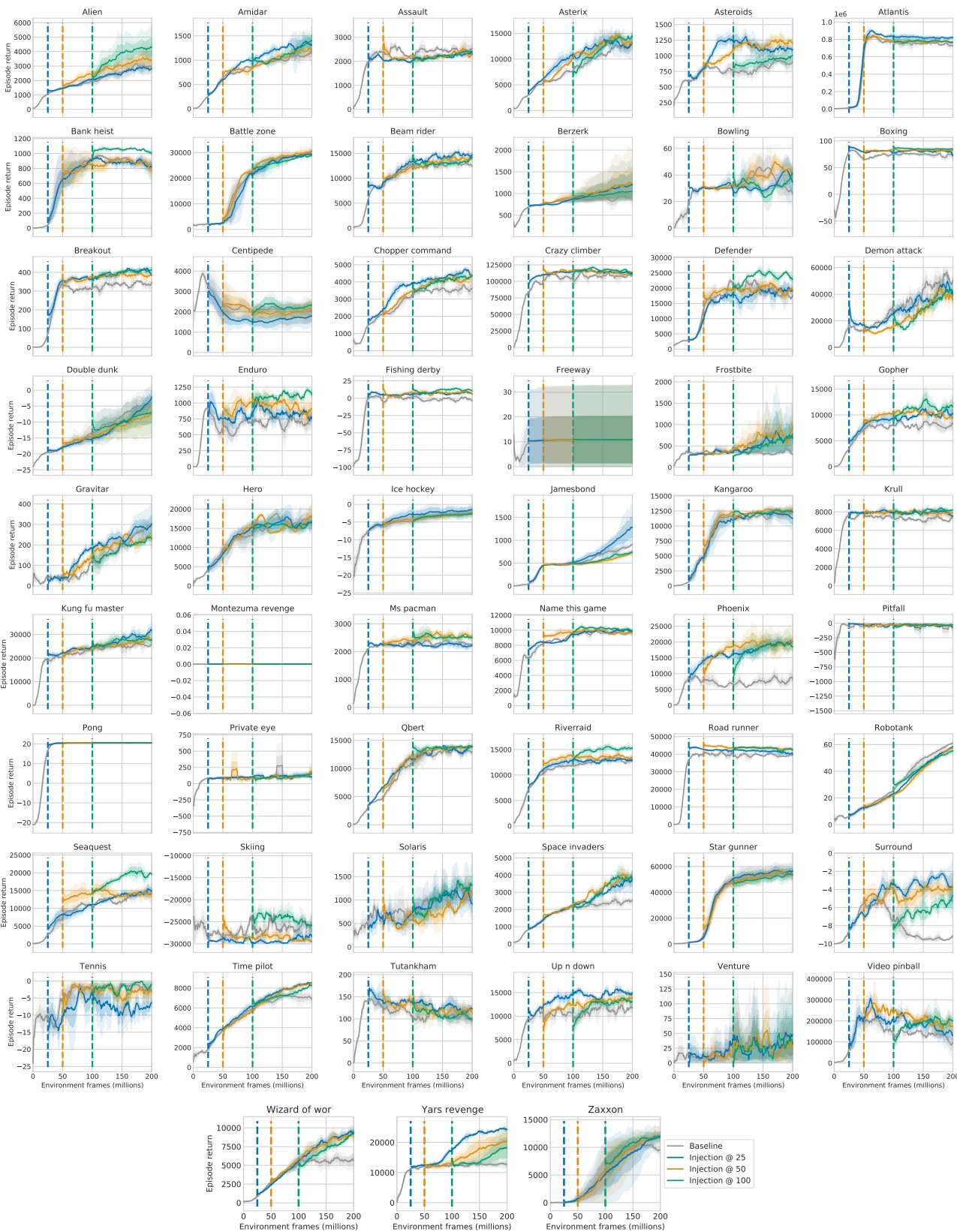

*Figure 9.* Performance of the Double DQN with and without plasticity injection after 25M, 50M, and 100M frames on the full Atari 57 benchmark. The potential discontinuities in the plots such as in `Road runner` are caused by the evaluation each 1M frames, i.e. the first moment the agent with injection contributes to the plot is after learning for 1M frames.

| Injection Effect | Environments |
|---|---|
| Consistent Improvement | `Alien`, `Asteroids`, `Breakout`, `Chopper command`, `Enduro`, `Frostbite`, `Gopher`, `Phoenix`, `Space invaders`, `Surround`, `Wizard of wor`, `Yars revenge` (12 total) |
| Minor Improvement | `Amidar`, `Asterix`, `Atlantis`, `Bank heist`, `Beam rider`, `Berzerk`, `Boxing`, `Defender`, `Fishing derby`, `Jamesbond`, `Krull`, `Ms pacman`, `Road runner`, `Seaquest`, `Time pilot`, `Up n down`, `Video pinball`, `Zaxxon` (18 total) |
| Negligible | `Battle zone`, `Bowling`, `Centipede`, `Crazy climber`, `Double dunk`, `Freeway`, `Gravitar`, `Hero`, `Ice hockey`, `Kangaroo`, `Kung fu master`, `Montezuma revenge`, `Name this game`, `Pitfall`, `Pong`, `Private eye`, `Qbert`, `Riverraid`, `Skiing`, `Solaris`, `Star gunner`, `Tennis`, `Tutankham`, `Venture` (24 total) |
| Negative | `Assault`, `Demon attack`, `Robotank` (3 total) |

*Table 1.* Summary of effects from applying plasticity injection to Double DQN agent on all 57 Atari games.

## A. Complete Learning Curves

Figure 9 presents the return plots over the course of Double DQN training for 200M frames on the whole set of 57 Atari games. We informally categorized environments into four buckets upon visual inspection of effects from plasticity injection in Table 1. The most notable negative example is `Demon attack`, while on `Assault` and `Robotank` the effect is negative but minor. In the rest of the 54 games, plasticity injection either improves performance or has a negligible effect, possibly depending on the injection timestep.

## B. Adaptive Criterion for Injection

As a step towards getting rid of the need to specify the injection timestep, we also explored the option of having a criterion for triggering the intervention. If the agent has the initial weight magnitude $\|w_0\|$ ($w$ denotes here both encoder and head weights), we inject plasticity after the weight norm surpasses the $3\|w_0\|$ threshold. The IQM scores of the agent with injection after 50M steps and with this simple heuristic coincide, although the frame when the agent reaches the threshold differs per game significantly: for some environments, it can be as small as 20M (such as `Enduro`), for other environments, it can be beyond 200M

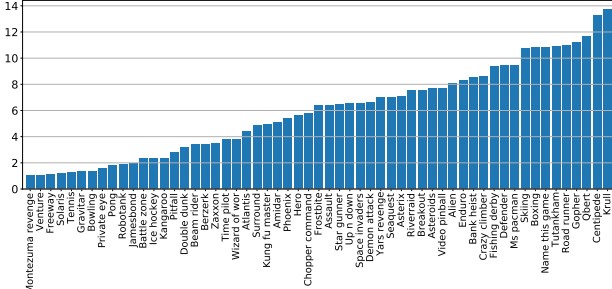

*Figure 10.* Per-game ratios of weight magnitude after learning for 200M frames and before experiencing any data. The ratios can vary up to 10 times between games.

(such as `Robotank`) implying that the agent will learn without injection. Figure 10 gives an overview of how much the weight norm grows over the course of training. We view devising an even more powerful criterion as a promising avenue for future work.

## C. On L2 Regularization.

The observations about the norm increase made us try adding L2 regularization to the Double DQN agent. A grid search over $[10^{-7}, 3 \cdot 10^{-7}, 10^{-6}, 3 \cdot 10^{-6}, 10^{-5}, 3 \cdot 10^{-5}]$ coefficients resulted in the best coefficient of $3 \cdot 10^{-6}$ but leaving the aggregate score mostly the same; higher values resulted in significant performance deterioration. The result gives evidence that controlling the weight norm itself does not address plasticity loss but allows multiple interpretations. We speculate that L2 might be prematurely encouraging weights to have zero magnitude before obtaining high rewards (the effect would be especially profound in sparse reward settings) or that L2 might have undesirable side effects of smoothing approximate value functions while the true value functions might be non-smooth (Dong et al., 2020). We are puzzled about the inefficacy of L2 in our experiments and mixed results from applying it in RL in past works: the majority of deep RL algorithms do not use it (Mnih et al., 2015; Schulman et al., 2015; Lillicrap et al., 2015; Mnih et al., 2016; Bellemare et al., 2017), although not without exceptions (Schrittwieser et al., 2021). Some works have explicitly reported negative effects from applying L2 in deep RL (Nikishin et al., 2022), while others highlighted its theoretical benefits (Farahmand et al., 2008); more research in needed to understand its effect in RL.

## D. Details about the Baselines

In Section 5.3, we considered three alternative ways of dynamically during training: resets, Shrink-and-Perturb (SnP), and naive width scaling. Resets re-initialize parameters of the last layers (using our notation, it corresponds to replacing $h_\theta(\cdot)$ with $h_{\theta'_1}(\cdot)$)) for given timesteps and rely on a replay buffer to transfer knowledge before and after the

intervention. Resets require the number of last layers specification and the application timestep. We ran a sweep over [1, 2] layers and two choices of timesteps: either once at 50M frames or trice at 50M, 100M, and 150M. Afterwards, we reported the results that attain the highest IQM score.

Shrink-and-Perturb modify all network weights $w$ as $w \leftarrow \lambda w + \sigma \epsilon$ at the given application timesteps, where $\epsilon$ is a random vector with the same dimensionality as $w$ sampled from the standard Gaussian distribution. SnP has three hyperparameters: the shrink coefficient $\lambda$, the noise scale $\sigma$, and the application timesteps. We performed a grid search over $\lambda$ in [0.1, 0.3, 1], $\sigma$ in [0.01, 0.1, 1], and the same choices of timesteps as for resets.

The best hyperparameters ended up being the ones that somewhat minimized the effect of both resets (1 layer, 1 application time) and SnP ($\lambda = 1$, $\sigma = 0.01$, 3 application times); other hyperparameters resulted in even worse performance. The paper on resets (Nikishin et al., 2022) demonstrates results on the Atari 100k benchmark (Kaiser et al., 2019) that focuses on a data-efficient regime with $10^5$ frames and contains a subset of 26 / 57 games. In this setting, the replay buffer has all experiences encountered during the agent's lifetime; this data can be sufficient for recovering the performance after a reset. In the Atari 200M setting though, the replay buffer has only 4M frames which might not be enough to recover fast after a reset. We speculate that similar reasoning applies to SnP since it can be seen as a soft version of resets (D'Oro et al., 2023).

For the width scaling method, we modify the last two layers by doubling their width. Suppose the weight matrices are $W_1 \in \mathbb{R}^{N \times K}$ and $W_2 \in \mathbb{R}^{K \times |\mathcal{A}|}$, where $|\mathcal{A}|$ is the action space dimensionality. We create two new matrices $W_1' \in \mathbb{R}^{N \times 2K}$ and $W_2' \in \mathbb{R}^{2K \times |\mathcal{A}|}$ and fill the first $K$ columns of $W_1'$ with values of $W_1$ and the first $K$ rows of $W_2'$ with values of $W_2$. The remaining entries are randomly set using the standard initializer. We perform a modification to the bias term $b_2' \in \mathbb{R}^{2K}$ by copying values from $b_2 \in \mathbb{R}^K$ and setting the rest to zero. The width is scaled once at 50M.

Such a naive approach increases plasticity but its inability to improve over the standard Double DQN might be attributed to adverse effects on the agent's predictions after the intervention without output correction.

## E. Computational Efficiency in Atari 200M

This appendix provides an example of how much computations can be saved with the dynamic growth of the network. The Double DQN agent from our codebase based on the open-source codebase (Quan & Ostrovski, 2020) takes about 6 days to learn in an Atari game for 200M frames using an A100 GPU. Figure 6 (right) demonstrates that an agent with plasticity injection after 50M frames does not compromise

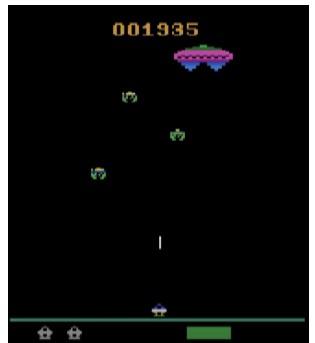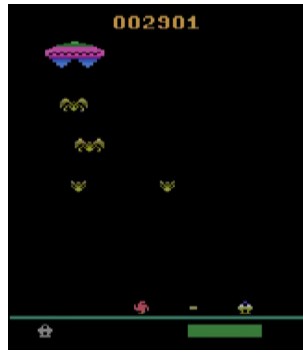

*Figure 11.* A demonstration of the `Assault` game evolution when a high-performing agent found on the Internet reaches a score of around 2800: before, the agent had to shoot only upwards; afterwards, it has to shoot up, left, and right. We interpret that the failure to improve upon the 2800 score is explained by exploration.

the aggregate performance compared to an agent with extra plasticity from the start. Hence the savings are occurring during the first 1.5 GPU-days. In our experiments, an agent with plasticity injection has about 15% lower training and evaluation speed, leading to savings of about 5.5 hours during the first 50M frames. Seemingly small in comparison to 6 days, training an agent in 57 games with 3 seeds takes 171 jobs, in total yielding savings of about 38 GPU-days. Such a difference can be non-trivial when computational resources are limited.

We emphasize that calculations here are an example; the GPU utilization in our experiments was around 10–20%, so it was not the biggest bottleneck. The amount of savings would depend on the implementation, hardware, an architecture, and the injection timestep, possibly resulting in larger computational efficiency in other domains.

## F. The `Assault` Game Analysis

We searched for a high-scoring behavior demonstration in the `Assault` environment on YouTube[5]. The screenshots in Figure 11 demonstrate the change of the environment around the score of 2800: before, the enemies were appearing only above the controlled starship, while afterwards, they start to appear from the left and from the right. Before the transition, the algorithm learned that actions "shoot left" and "shoot right" were irrelevant, while afterwards, it has to start using these actions, suggesting that the performance plateau can be attributed to exploration challenges.

We highlight that it was the suggested protocol for diagnosis that led to the insight: after seeing that the post-injection agent has the same performance plateau as the baseline, we decided to investigate the behavior in the game and realized that previously irrelevant actions became critical.

---

[5]https://youtu.be/HwWJrb2PQQ0