# OpenReview forum: "Deep Reinforcement Learning with Plasticity Injection"
_ICLR.cc/2023/Workshop/RRL — RRL 2023 Spotlight_

### Official Review · Reviewer_7JWB · 2023-02-23
**Simple way to test plasticity with thorough empirical evaluation**

**Rating:** 4
**Confidence:** 4

**Review:**

Strengths:
1. The paper presents an simple and elegant way to test plasticity in the neural networks. They accomplish this by freezing their original neural network and adding two function (neural networks) whose sum is zero initially and one of them is trainable. Thus, we have a increases capacity neural network with same number of trainable variables with the same function value initially at all points in the input domain.
2. The paper is able to detach the effect of exploration from plasticity by testing empirically if the performance during training is affected during plasticity injection.
3. The empirical evaluation is thorough on the Atari learning environment which clearly shows that loss of plasticity can be identified, and increased plasticity through their method could make the RL training more performant and even memory efficient.
4. The ablations have done a good effort to remove any other potential sources of changes introduced due to their plasticity injections that could be confounding their analysis.
5. The paper is quite relevant to the workshop as it directly addresses the question of reincarnating RL.

---

### Official Review · Reviewer_LR9h · 2023-02-27
**Intriguing study of plasticity in deep reinforcement learning models, lots of interesting experiments, ablation studies and conclusions.**

**Rating:** 4
**Confidence:** 4

**Review:**

Overall evaluation:
* The paper fits perfectly within the framework of this workshop as it investigates how the plasticity of a RL model changes throughout the training, which is necessary for reusing it further.
* The paper is very clearly written. The experiments are well-motivated, the chain of thought of the authors is easy to follow and the appendix contains a lot of useful details.
* The technical quality of the paper is quite high, especially for a workshop paper. The authors run experiments on the whole Atari 57 benchmark, use multiple seeds and apply good practices in terms of looking out for statistical significance.
* The results are of high interest to the community.

As such, I think this paper is a very good fit for this venue. Below, I attached some comments and suggestions, but I still think the paper is very good.

Feedback and comments:
* The results in Figure 6, right worry me a bit. That is, looking at the "Unfrozen injection @ 0", it seems that simply having two "heads"  solves the problem. Granted, you save some computation if you apply injection after some time, but it seems that the plasticity loss here is not a fundamental issue, the network architecture was wrong. As you say, and I agree, "RL agents typically employ networks that were originally proposed for stationary problems, but perhaps a specialized parameterization would suit the non-stationary nature of RL better". But it seems that the specialized parameterization seems to be the crucial point here, not the plasticity loss during the training - you can just change it before you train and it works just as well. Also, I wonder if having two heads is enough or if we need the $\theta'_1, \theta'_2$ distinction from Equation 1.
* I would be interested to see the same experiments in some other domains. For example, do your observations transfer to continuous control problems such as DMControl or Mujoco? In particular, I wonder how important is the image encoding - can we see the same or "raw" states?
* I would appreciate a more thorough comparison to Progressive Networks since that method seems very similar to the proposed plasticity injection. Obviously, there are a few things that are different (randomly initialized columns, etc), but the two ideas seem deeply connected.
* The plasticity loss experiment in Figure 1 is quite striking, especially since it technically does not include any kind of RL, just simple MSE minimization in a continual learning setting. I think more details about this experiment should be included. What was the average return of the RL policy used for this experiment? What was the architecture used? Do we see the same behaviour for other tasks? Similar experiments were run previously in the Ash & Adams and Beriaru et al. papers you cite, but I don't think the results were quite as striking. It would be quite interesting to pinpoint the exact reason for this plasticity loss. In particular [1] shows that the input masking in RL impacts the performance quite a lot. One possible hypothesis would be that some of the neurons corresponding to certain inputs activations "die" in the early phase of the training when these pixels are not important. And later, when these start being important, we can no longer activate them. I'm not saying that is the correct solution here, but a more in-depth investigation would be quite valuable for the community.
* I don't think the code was shared, and I think it would be interesting to tinker with it.
* I don't think the exact network architecture is mentioned anywhere explicitly. I assume you use the one from the Double DQN paper, but I think it's quite an important point of the paper and as such it would be useful to write it down explicitly (size of the convolutional layers, the width of the head, etc).

[1] On Lottery Tickets and Minimal Task Representations in Deep Reinforcement Learning, Marc Aurel Vischer*, Robert Tjarko Lange*, Henning Sprekeler, ICLR 2022